Methods

# MagnEdit—interacting factors that recruit DNA-editing enzymes to single base targets

Jennifer L McCann[1,2,3,4,5], Daniel J Salamango[1,2,3,4], Emily K Law[1,2,3,4,5], William L Brown[1,2,3,4] , Reuben S Harris[1,2,3,4,5]

**Although CRISPR/Cas9 technology has created a renaissance in genome engineering, particularly for gene knockout generation, methods to introduce precise single base changes are also highly desirable. The covalent fusion of a DNA-editing enzyme such as APOBEC to a Cas9 nickase complex has heightened hopes for such precision genome engineering. However, current cytosine base editors are prone to undesirable off-target mutations, including, most frequently, target-adjacent mutations. Here, we report a method to "attract" the DNA deaminase, APOBEC3B, to a target cytosine base for specific editing with minimal damage to adjacent cytosine bases. The key to this system is fusing an APOBEC-interacting protein (not APOBEC itself) to Cas9n, which attracts nuclear APOBEC3B transiently to the target site for editing. Several APOBEC3B interactors were tested and one, hnRNPUL1, demonstrated proof-of-concept with successful C-to-T editing of episomal and chromosomal substrates and lower frequencies of target-adjacent events.**

## Introduction

The original BE3 cytosine base editor (CBE) comprised the rat APOBEC1 deaminase fused to the N-terminal end of a Cas9 nickase (Cas9n D10A (1)). Appropriate gRNAs are able to target this assembly to genomic cytosine bases and facilitate high-frequency editing (10–90% depending on a number of variables including distance between target cytosine and protospacer adjacent motif) (1, 2). However, this technology is prone to a number of off-target effects, including RNA editing (3, 4), random genomic DNA editing (5, 6, 7, 8), and most frequently target-adjacent editing (1, 2, 5, 9, 10). The latter problem is due predominantly, if not exclusively, to deamination of single-stranded DNA cytosines located adjacent to the desired target cytosine in the same gRNA-displaced R-loop. This issue has been diminished—but not eliminated—by mutating APOBEC1 (3, 4, 10, 11), trying different DNA deaminase family members (12, 13, 14, 15,

16, 17, 18), mutating Cas9 (10, 19, 20, 21, 22, 23, 24), and leveraging different Cas enzymes (11, 16, 24, 25). However, an invariant feature of almost all current designs is *covalent* fusion of the deaminase to the Cas9 complex, which traps the tethered deaminase locally and inextricably links both on-target (desirable) and target-adjacent (undesirable) cytosine deamination events (schematic in Fig 1A).

We hypothesize that *non-covalent* methods to "attract" a DNA cytosine deaminase to a particular genomic cytosine target will decouple the fates of on-target and target-adjacent editing events and thereby enhance the likelihood of achieving precise single base substitution mutations. A key to implementing this non-covalent strategy is identifying appropriate APOBEC-interacting proteins, which bind the deaminase without blocking the active site from engaging a target cytosine. Such interacting proteins can then be tethered to a Cas9n/gRNA complex and used to "attract" a co-expressed APOBEC enzyme (exogenous or endogenous) to edit a particular genomic target cytosine. Inspired by the analogy to magnetism, this system is called MagnEdit (schematic in Fig 1B).

## Results

### Covalent CBE versus non-covalent MagnEdit technology for DNA cytosine base editing

As an initial test of MagnEdit, we fused APOBEC3B (A3B)-interacting proteins from the literature (simian immunodeficiency virus [SIV] Vif (26), hnRNPK (27)) and proteomic screens (CDK4 (28) and McCann et al, unpublished) to the N-terminal end of Cas9n and asked whether these complexes are able to recruit A3B to edit an episomal *eGFP* reporter (13) in 293T cells (TC to TT schematic in Fig 1B and actual *eGFP* gRNA target sequence in Fig 1C inset). Because of simultaneous overexpression of reaction components following co-transfection, including A3B, a low level of eGFP-positive cells (~1–2%) was observed in the absence of a gRNA and a candidate interacting protein (reactions represented by gray and black bars in Fig 1C). Interestingly, addition of *eGFP* Leu202-targeting gRNA (again without an interactor) enabled higher levels of eGFP editing by A3B

[1]Department of Biochemistry, Molecular Biology and Biophysics, University of Minnesota, Minneapolis, MN, USA    [2]Institute for Molecular Virology, University of Minnesota, Minneapolis, MN, USA    [3]Masonic Cancer Center, University of Minnesota, Minneapolis, MN, USA    [4]Center for Genome Engineering, University of Minnesota, Minneapolis, MN, USA    [5]Howard Hughes Medical Institute, University of Minnesota, Minneapolis, MN, USA

Correspondence: rsh@umn.edu

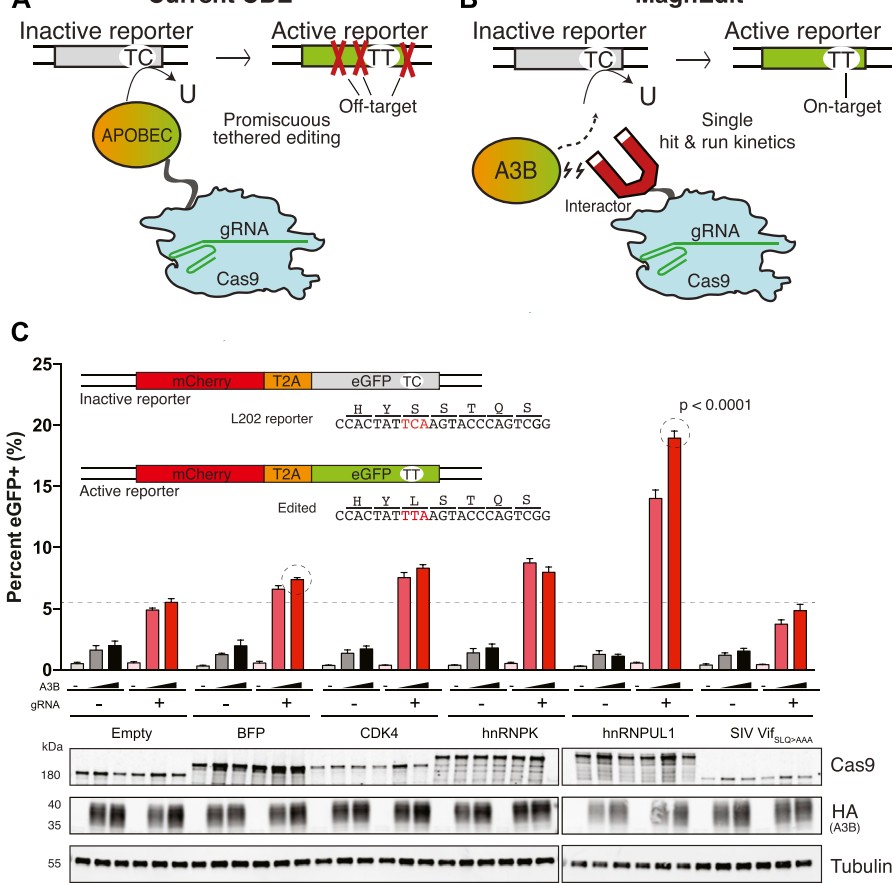

**Figure 1. Covalent CBE versus non-covalent MagnEdit technology for DNA cytosine base editing.**
**(A)** Schematic of current CBE methodology with APOBEC-Cas9n/gRNA editosome engaging the *eGFP* Leu202 reporter. Target-adjacent mutations are indicated by red X's. **(B)** Schematic of MagnEdit with interactor–Cas9n/gRNA complex recruiting untethered A3B to the *eGFP* Leu202 reporter. **(C)** Quantification of episomal eGFP reporter editing activity of the indicated MagnEdit complexes in 293T cells (n = 3 biologically independent experiments, average ± SD, P < 0.0001 by unpaired *t* test for circled reactions). The immunoblots below are from one of these experiments. The inset schematic shows the *eGFP* Leu202 reporter, the DNA region matching the gRNA, and the target cytosine in red.

(~5–7%; empty-Cas9n plus gRNA reaction in Fig 1C). Unfortunately, most MagnEdit complexes failed to stimulate editing beyond these background levels or those caused by a non-interacting blue fluorescent protein (BFP)-Cas9n control (Fig 1C). SIV Vif (SLQ-AAA)-Cas9n even yielded lower overall frequencies of background editing, likely because of poorer expression relative to other MagnEdit constructs (the SLQ-AAA was necessary to prevent Vif from binding ELOC and triggering A3B degradation (26)). However, one MagnEdit construct, hnRNPUL1-Cas9n, was clearly capable of recruiting A3B in a dose-dependent manner to catalyze editing and activation of the *eGFP* reporter (Fig 1C). Editing frequencies due to hnRNPUL1-Cas9n were at least twofold higher than the BFP-Cas9n/gRNA-induced background in these transient transfection experiments (P < 0.0001 by unpaired *t* test).

## Chromosomal DNA editing by MagnEdit

Next, we analyzed chromosomal DNA editing by MagnEdit. The same *eGFP* Leu202 reporter was integrated into the genome of 293T cells by low MOI lentiviral transduction followed by hygromycin selection to ensure that every cell has one editing target (uniform mCherry-positive population confirmed by flow cytometry). This pool was then transfected, as above, with the panel of A3B interactor–Cas9n complexes with or without the Leu202-targeting gRNA in the presence or absence of exogenous A3B. Also, as above, empty-Cas9n and BFP-

Cas9n were used as negative controls, and most MagnEdit complexes showed no activity above background levels. Flow cytometry noise was the likely source of these low background levels of eGFP positivity because no difference was seen here with/without the *eGFP* Leu202-targeting gRNA or different amounts of A3B. However, in agreement with episomal editing data, hnRNPUL1 MagnEdit complexes yielded a dose-dependent increase in A3B editing (quantification and representative immunoblots in Fig 2A; P < 0.0009 by unpaired *t* test). As expected, all components of the MagnEdit reaction were required for chromosomal DNA editing (hnRNPUL1-Cas9n complex, Leu202 gRNA, and A3B-HA; Fig 2B).

To further investigate the mechanistic requirements for MagnEdit, we asked whether the nuclear import activity of A3B is required. A3B is the only constitutively nuclear human APOBEC family member (29, 30, 31) and nuclear localization is predicted to be essential for MagnEdit activity. Recent studies have combined to delineate a non-canonical nuclear import mechanism involving multiple A3B surface residues in two distinct patches (31). Indeed, two previously characterized import-defective mutants of A3B, V54D (29), and chimera 22-32 (31) were no longer capable of editing the chromosomal *eGFP* Leu202 reporter (Fig 2C). These amino acid substitutions localize to the N-terminal regulatory domain of A3B and the editing phenotype is indistinguishable from that of a C-terminal domain catalytic mutant (CM in Fig 2C). In addition, the

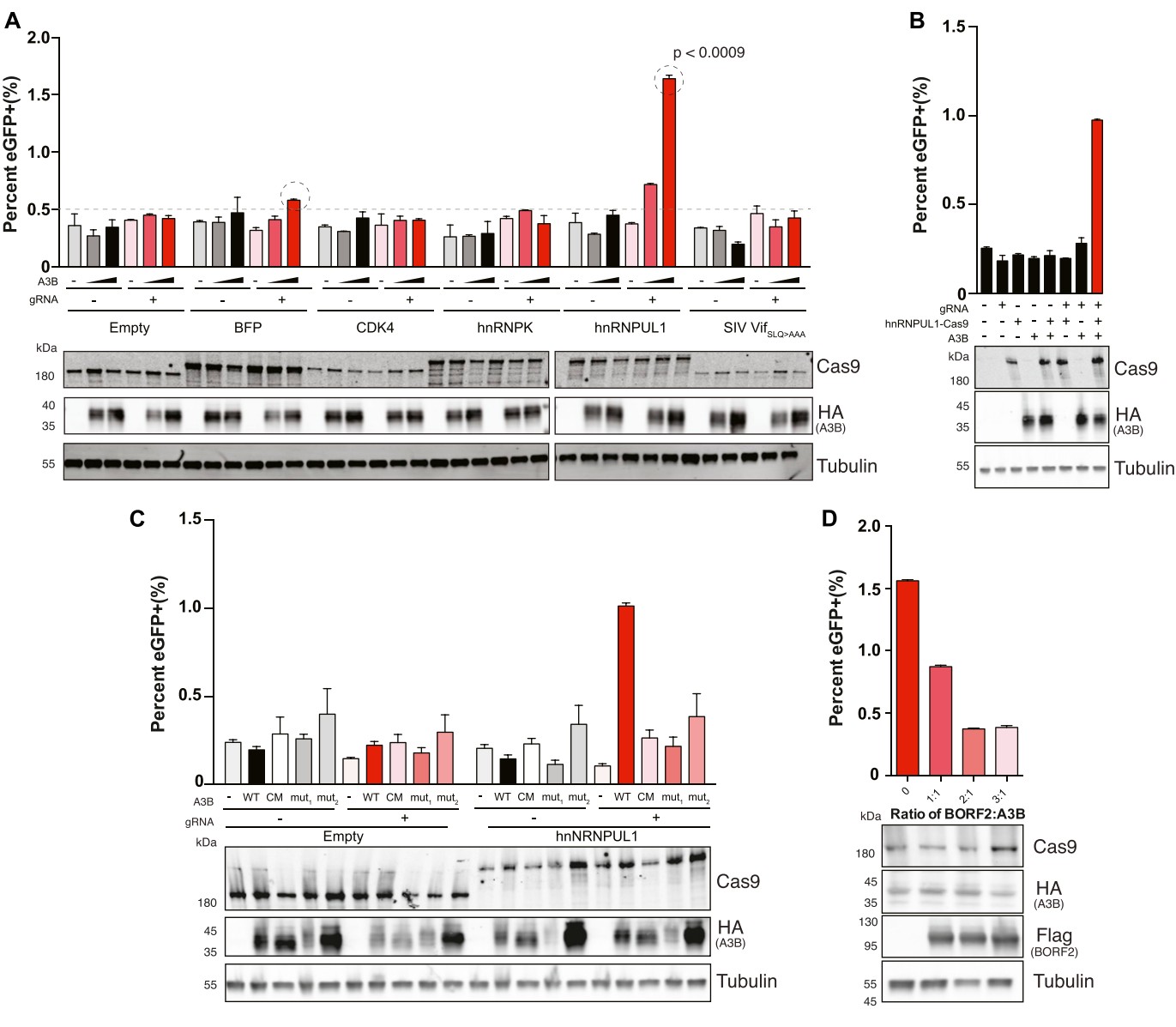

**Figure 2. Chromosomal DNA editing by MagnEdit.**
**(A)** Quantification of chromosomal eGFP reporter editing activity of the indicated MagnEdit complexes in 293T cells (n = 3 biologically independent experiments, average ± SD, $P < 0.0009$ by unpaired $t$ test for circled reactions). The immunoblots below are from one of these experiments. **(B, C, D)** Chromosomal eGFP editing activity for reactions containing the indicated components (n = 3, average ± SD). The immunoblots below each histogram are from one of the experiments.

chromosomal DNA-editing reaction can be suppressed in a dose-dependent manner by BORF2, a recently discovered A3B antagonist encoded by Epstein-Barr virus (32) (Fig 2D).

## L202 reporter editing by CBE versus MagnEdit

DNA sequencing was used to compare the ratios of on-target and target-adjacent editing by a current CBE (A3B-Cas9n) (13) and the MagnEdit complex described here (A3B plus hnRNPUL1-Cas9n). A3B-Cas9n was used for these comparisons because its catalytic domain is less promiscuous than BE3 (13) and it provides an isogenic comparison for covalent versus non-covalent editing reactions catalyzed by A3B. As above, chromosomal DNA editing was performed by transfecting Cherry-

positive 293T pools with the *eGFP* Leu202 gRNA expression vector and plasmids encoding either A3B-Cas9n or hnRNPUL1-Cas9n with a separate vector for A3B. FACS was used 72 h post-transfection to isolate eGFP-positive pools for target recovery and deep sequencing. As indicated by bright eGFP-positive signals in each editing reaction 72 h post-transfection, both editing technologies activated the reporter with the A3B CBE appearing approximately fourfold more efficient (6.1% for A3B-Cas9n and 1.5% for A3B plus hnRNPUL1-Cas9n; Fig 3A). In each instance, FACS resulted in enrichment of similar numbers of eGFP-positive cells for deep sequencing of the Leu202 target codon and flanking DNA (98% for A3B-Cas9n and 99% for A3B plus hnRNPUL1-Cas9n; Fig 3A).

As negative controls, parallel reactions without gRNAs were directly converted to genomic DNA for deep sequencing and no

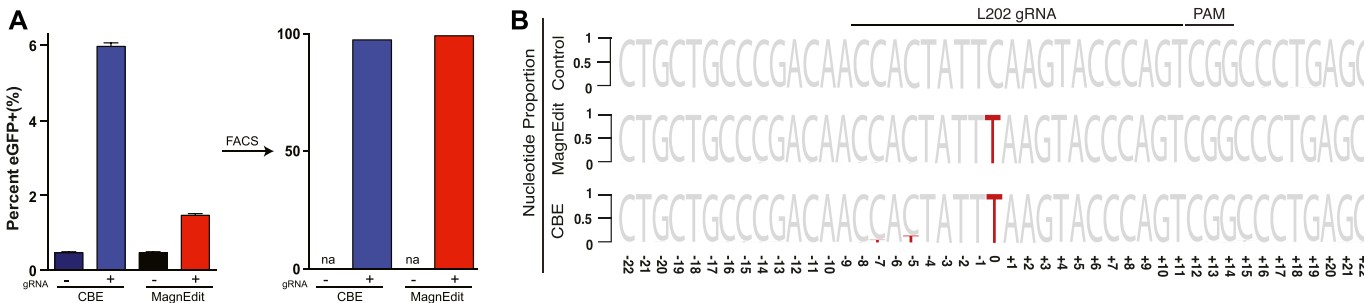

**Figure 3. Target-adjacent editing by CBE versus MagnEdit.**
**(A)** Quantification of eGFP-positive 293T cells (*eGFP* Leu202 edited) post-editing and pre-/post-enrichment by FACS for the indicated editing reactions (n = 3 technical replicate experiments, average ± SD). **(A, B)** Sequence logos summarizing MiSeq data from the same reactions as panel (A). The consensus sequence matches the single-stranded DNA region displaced by gRNA annealing with the target cytosine. Red coloring highlights mutations that occurred in >5% of the MiSeq reads for each reaction (numbers are nucleobase distances 5′ or 3′ of the target "C").

target cytosine mutations were observed. As anticipated above and from prior studies (13), the inclusion of a gRNA enabled both technologies to restore functionality to *eGFP* codon 202 (TCA [Ser] to TTA [Leu]; represented by a red T and normalized to 1 for comparisons in Fig 3B). However, target-adjacent editing frequencies were clearly different for these two different base editing technologies. The covalently tethered A3B-Cas9n CBE caused high frequencies of target-adjacent editing within the R-loop created by gRNA binding (14.8% at the −5 position and 6.4% at the −7 position in Fig 3B). In contrast, the hnRNPUL1-Cas9n MagnEdit system showed lower target-adjacent editing within the gRNA-binding region (2.5% at both −5 and −7 positions in Fig 3B).

### Chromosomal DNA editing by CBE versus MagnEdit

To further investigate the accuracy of the MagnEdit system, we compared ratios of on-target and target-adjacent editing by a current CBE (A3B-Cas9n) (13) and the MagnEdit complex described here (A3B plus hnRNPUL1-Cas9n) at two genomic loci, *FANCF* and *EMX1*, reported previously (1). As above, chromosomal DNA editing was performed by transfecting Cherry-positive 293T pools with gRNAs targeting both the *eGFP* Leu202 reporter and *FANCF* or *EMX1* and plasmids encoding either A3B-Cas9n or hnRNPUL1-Cas9n with a separate vector for A3B. FACS was used 72 h post-transfection to isolate eGFP-positive pools for target DNA recovery and deep sequencing. Similar to results in Fig 3A, both editing technologies activated the *eGFP* reporter with, again, the A3B CBE appearing approximately fourfold more efficient (Fig 4A and E).

As negative controls, parallel reactions without gRNAs were directly converted to genomic DNA for deep sequencing and no target cytosine mutations were observed in *FANCF* or *EMX1* (control reactions in Fig 4B and F). However, upon inclusion of appropriate gRNAs targeting these genes, clear differences in accuracy were observed between these two different base editing technologies. Similar to prior literature for *FANCF* editing by BE3 (1), the covalently tethered A3B-Cas9n CBE caused high frequencies of target-adjacent editing within the R-loop created by gRNA binding (42% at the +1 position and 35% at the +2 position in Fig 4B). It also caused significant off-target editing at the −9 position, which is just upstream of the gRNA-binding region (13.9% in Fig 4B). In contrast, the hnRNPUL1-Cas9n MagnEdit system showed significantly lower target-adjacent editing within the gRNA-binding region and no detectable editing outside of the gRNA-binding region (13% at the +1

position, 20% at the +2 position, and 0.5% at the −9 position in Fig 4B). Although target-adjacent editing is higher in *FANCF* than in the *eGFP* L202 reporter, this is likely due to the trinucleotide context of *FANCF* being "TCC" rather than "TCA" (i.e., TCC is a suboptimal context for A3B as shown by biochemical and structural studies (33)). Nevertheless, upon consideration of all possible editing permutations within the gRNA-binding region (on-target and target-adjacent events), the hnRNPUL1-Cas9n MagnEdit system shows a twofold increase in on-target editing in comparison to the covalently tethered A3B-Cas9n CBE (19% versus 9% in Fig 4C and D, respectively). The hnRNPUL1-Cas9n MagnEdit system yields correspondingly fewer target-adjacent editing events than the A3B-Cas9n CBE system (21.8% versus 45.5% in Fig 4C and D, respectively).

Similar trends were evident for the chromosomal *EMX1* locus. The covalently tethered A3B-Cas9n CBE caused high frequencies of target-adjacent editing within the R-loop created by the gRNA binding (58.5% at the +1 position in Fig 4F). In contrast, the hnRNPUL1-Cas9n MagnEdit system showed more than threefold lower target-adjacent editing within the gRNA-binding region (15.0% at the +1 position in Fig 4F). Again, this genomic target has a trinucleotide context of "TCC" rather than "TCA," so editing results were broken down into trinucleotide contexts for further consideration. The hnRNPUL1-Cas9n MagnEdit system specifically edited the target "C," whereas the covalently tethered A3B-Cas9n CBE was less specific (49% versus 18.2% on-target editing, respectively, Fig 4G and H). These results combine to demonstrate that the MagnEdit system yields higher frequencies of on-target editing with significantly lower frequencies of target-adjacent editing events. In addition, higher *FANCF* and *EMX1* on-target editing frequencies and similar adjacent off-target trends were evident for MagnEdit versus the covalently tethered A3B-Cas9n CBE in eGFP-negative pools (Fig S1). These additional results from sequencing the "dark" population suggested that on-target chromosomal editing events may far exceed those that yield functional correction of the *eGFP* Leu202 reporter.

## Discussion

This study describes a fundamentally different approach to single base editing through the use of non-covalent interactions to "attract" a DNA cytosine deaminase to a single target cytosine. A3B is

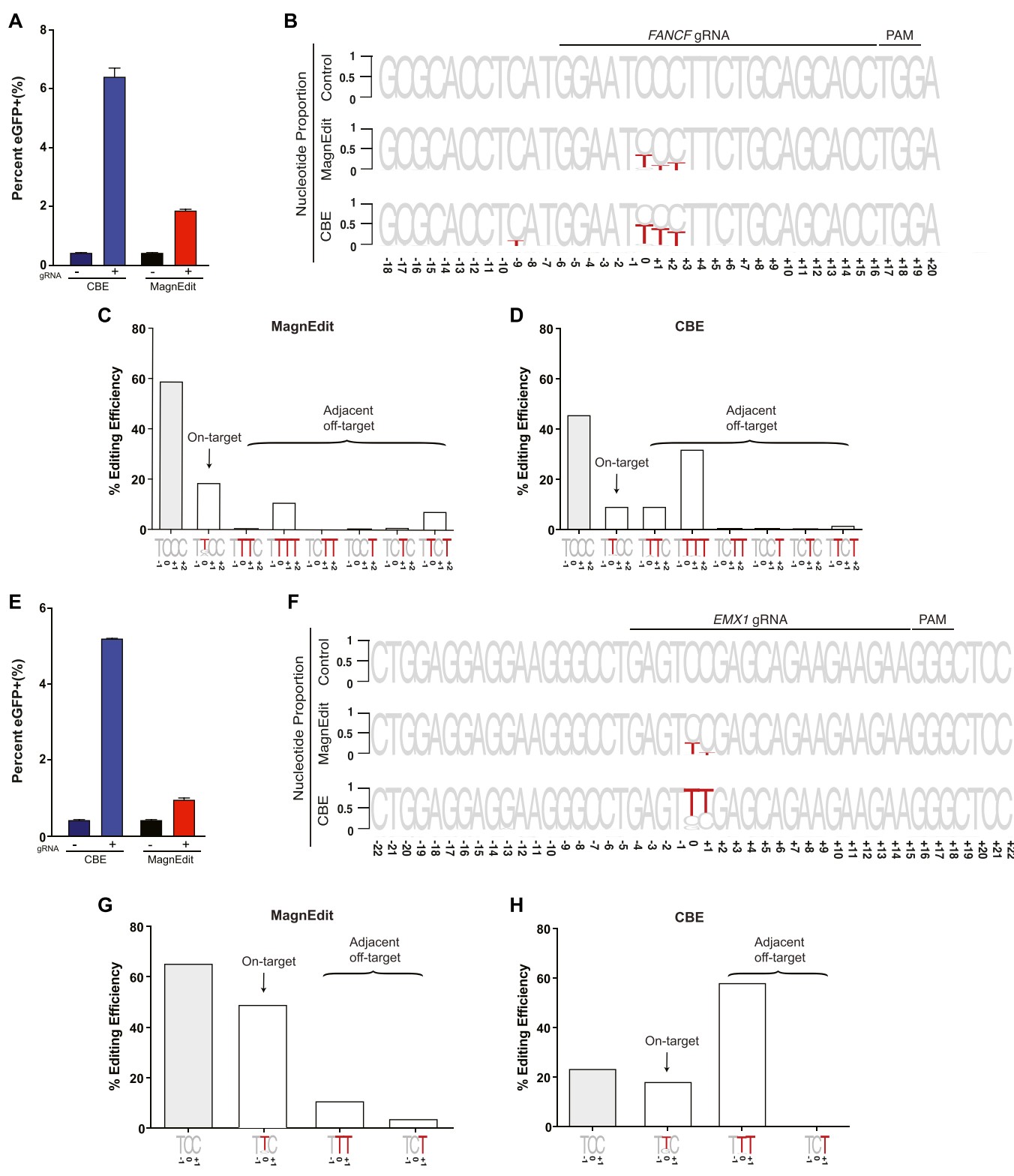

**Figure 4. Chromosomal DNA editing by cytosine base editor (CBE) versus MagnEdit.**
**(A)** Quantification of eGFP-positive 293T cells (*eGFP* Leu202 edited with co-delivery of *FANCF* gRNA) post-editing and pre-enrichment by FACS for the indicated editing reactions (n = 3 technical replicate experiments, average ± SD). **(A, B)** Sequence logos summarizing MiSeq data of *FANCF* from the same reactions as panel (A). The consensus sequence matches the single-stranded DNA region displaced by gRNA annealing with the target cytosine. Red coloring highlights base substitution mutations that occurred in >5% of the MiSeq reads for each reaction (numbers are nucleobase distances 5' or 3' of the target "C"). **(B, C)** Quantification of single nucleobase substitution mutations from the MagnEdit reaction shown in panel (B). **(B, D)** Quantification of single nucleobase substitution mutations from the CBE reaction shown in

desirable for this application because it is normally nuclear (not shuttling or cytoplasmic like related family members) (29, 30, 31, 34, 35) and, because of active site structural constraints (33, 36, 37), unlikely to elicit RNA level off-target editing events as documented recently for BE3 and A3A CBEs (3, 4). A variety of techniques may be used in the future to identify additional APOBEC-interacting "baits" for the MagnEdit system (proteomic, genetic, and directed-evolution,). Naturally or artificially engineered antibodies may also be effective. Similar MagnEdit approaches may also benefit adenosine base editing systems. Although we have not observed nor found literature indicating potential side effects of overexpressing hnRNPUL1 in cells, heterologous attractants such as single-chain antibodies may be even better for promoting the non-covalent editing of single target cytosine bases.

In general, proteins such as hnRNPUL1 that interact with the non-catalytic N-terminal domain of A3B may be more effective than those that bind the catalytic C-terminal domain simply because they are less likely to interfere with catalytic activity. For instance, EBV BORF2 is the only A3B catalytic domain interactor described till date (32) and, as shown here, it potently blocks editing in the MagnEdit system. However, not all A3B-interacting proteins are likely to be effective in the MagnEdit system because affinities may be too low, nuclear access may not be allowed, and/or binding confirmations may be unproductive (e.g., CDK4, hnRNPK, and SIV Vif). Although others have used non-covalent methods such as SunTagging (38) and MS2 (18) to recruit DNA methyltransferase 3A (DNMT3A) and AID, respectively, these methods rely upon very high-affinity binding for positive results (39, 40). It is likely the high on-target and low target-adjacent editing of the MagnEdit system is due to an optimal affinity between hnRNPUL1 and A3B, which leaves A3B untethered and able to generate greater than random "hit-and-run" kinetics at the target "C." Methods such as SunTagging and MS2 may be efficient for editing enzyme recruitment but are unlikely to differ from covalent tethering in terms of increased accuracy due to high on-rates and low off-rates. Moreover, with additional optimization, it is likely that the MagnEdit system may be used with endogenous A3B and/or related A3 enzymes and, therefore, mitigate risks associated with overexpressing enzymes exogenously. Overall, the non-covalent MagnEdit system is attractive for helping to minimize off-target effects and ultimately enable true single base editing even though higher fidelities may come at a cost of lower overall efficiencies.

# Materials and Methods

## Cell lines

293T and 293T-Leu202 cells were cultured in RPMI 1640 supplemented with 10% FBS and penicillin–streptomycin. The chromosomal 293T-Leu202 reporter line was constructed using viral transduction followed by hygromycin selection (detailed below).

## Constructs

The rat APOBEC1-XTEN-Cas9n-UGI-NLS construct (BE3) was provided by David Liu (1). Interactor cDNA sequences were cloned into the BE3 vector in place of APOBEC1 using standard PCR subcloning techniques, thereby creating CMV-driven constructs with gene-of-interest followed by an XTEN linker to Cas9n (D10A)-UGI-NLS. GenBank accession numbers for BFP (MK178577.1), CDK4 (NM_000075.4), hnRNPK (NM_031263.4), and hnRNPUL1 (EU831487.1). SIV Vif was subcloned from a reported construct (26, 41). Leu202 gRNA, NS gRNA, empty-Cas9n-UGI-NLS, and Leu202 reporter (pLenti-CMV-mCherry-T2A-eGFP) have been reported (13). EMX1 and FANCF guide constructs were cloned into LentiCRISPR1000 (42) via Golden Gate cloning using the Esp3I sites. EMX1 and FANCF target sequences have been reported (1). All constructs were confirmed by Sanger DNA sequencing (GeneWiz). pcDNA3.1-3xHA, A3Bi-3xHA, and A3Bi$_{V54D}$-3xHA have been reported (29), and A3B$_{chim22-32}$-3xHA was subcloned from a previously reported construct (31). BORF2-3xFlag has also been reported (32).

## Episomal base editing experiments

Semi-confluent 293T cells in a six-well plate format were transfected with 200 ng gRNA, 400 ng reporter, 600 ng Cas9n-UGI-NLS, and either 600 ng pcDNA3.1-3xHA, 300 ng pcDNA3.1-3xHA, and 300 ng A3B-3xHA or 600 ng A3B-3xHA (25 min at RT with 3:1 ratio of TransIT LT1 [Mirus] and 250 μl of serum-free RPMI 1640 [Hyclone]). The cells were harvested after 72 h of incubation for editing quantification by flow cytometry.

## Chromosomal base editing experiments

Semi-confluent 10-cm plates of 293T cells were transfected with 8 μg of an HIV-1 Gag-Pol packaging plasmid, 1.5 μg of a VSV-G expression plasmid, and 3 μg of pLenti-CMV-mCherry-T2A-eGFP$_{Leu202}$-IRES-Hygro. Viruses were harvested 48 h post-transfection and used to transduce target cells. 48 h post-transduction, the cells were selected using 250 μg/ml hygromycin. Transduced, mCherry-positive cells were transfected with 600 ng Cas9n-UGI editor, 200 ng of Leu202 or NS gRNA, and either 600 ng pcDNA3.1-3xHA, 300 ng pcDNA3.1-3xHA, and 300 ng A3B-3xHA or 600 ng A3B-3xHA. The cells were harvested 72 h post-transfection, and editing was quantified by flow cytometry (fraction of eGFP and mCherry double-positive cells in the total mCherry-positive population). For EMX1 and FANCF targets, transduced, mCherry-positive cells were transfected with 600 ng Cas9n-UGI editor, 200 ng of Leu202 gRNA, and 200 ng of EMX1 or FANCF gRNA or just 200 ng of the NS gRNA and 600 ng A3B-3xHA. The cells were harvested 72 h post-transfection, and editing was quantified by flow cytometry (fraction of eGFP and mCherry double-positive cells in the total mCherry-positive population). FACS was used to collect both the

---

panel (B). **(E)** Quantification of eGFP-positive 293T cells (*eGFP* Leu202 edited with co-delivery of *EMX1* gRNA) post-editing and pre-enrichment by FACS for the indicated editing reactions (n = 3 technical replicate experiments, average ± SD). **(E, F)** Sequence logos summarizing MiSeq data of *EMX1* from the same reactions as panel (E). The consensus sequence matches the single-stranded DNA region displaced by gRNA annealing with the target cytosine. Red coloring highlights base substitution mutations that occurred in >5% of the MiSeq reads for each reaction (numbers are nucleobase distances 5′ or 3′ of the target "C"). **(F, G)** Quantification of single nucleobase substitution mutations from the MagnEdit reaction shown in panel (F). **(F, H)** Quantification of single nucleobase substitution mutations from the CBE reaction shown in panel (F).

mCherry-positive and the eGFP and mCherry double-positive cells for genomic DNA isolation and MiSeq analysis.

### MiSeq

*eGFP* target sequences were amplified using Phusion high-fidelity DNA polymerase (New England Biolabs) and previously reported primers (13). To add diversity to the sequence library, zero, one, or two extra cytosine bases were added to the 5′ end of the forward and reverse primers for each amplicon. Barcodes were added to generate full-length Illumina amplicons. Samples were analyzed using Illumina MiSeq 2 × 150-nucleotide paired-end reads (University of Minnesota Genomics Center). Reads were paired using FLASh (43). Data processing was performed using a locally installed FASTX-Toolkit. Fastx-clipper was used to trim the 3′ constant adapter region from sequences, and a stand-alone script was used to trim 5′ constant regions. Trimmed sequences were then filtered for high-quality reads using the Fastx-quality filter. Sequences with a Phred quality score less than 30 (99.9% base calling accuracy) at any position were eliminated. Preprocessed sequences were then further analyzed using the FASTAptamer toolkit (44). FASTAptamer-Count was used to determine the number of times each sequence was sampled from the population. Each sequence was then ranked and sorted based on overall abundance, normalized to the total number of reads in each population, and directed into FASTAptamer-Enrich. FASTAptamer-Enrich calculates the fold enrichment ratios from a starting population to a selected population by using the normalized reads-per-million (RPM) values for each sequence. For *eGFP* Leu202 reporter and edited sequence comparison, sequences over 5 RPM were included and only on-target sequences were used for Fig 3B. For *EMX1* and *FANCF* analysis, all sequences over 5 RPM were included (Fig 4).

### Immunoblots

$1 \times 10^6$ cells were lysed directly into 2.5× Laemmli sample buffer, separated by 4–20% SDS–PAGE, and transferred to PVDF-FL membranes (Millipore). The membranes were blocked in 5% milk in PBS and incubated with a primary antibody diluted in 5% milk in PBS supplemented with 0.1% Tween 20. Secondary antibodies were diluted in 5% milk in PBS supplemented with 0.1% Tween 20 and 0.01% SDS. The membranes were imaged with an LI-COR Odyssey instrument. Primary antibodies used in these experiments were rabbit anti-Cas9 (ab189380; Abcam), mouse anti-tubulin (T5168; Sigma-Aldrich), rabbit anti-HA (3724S; Cell Signaling), and mouse anti-Flag (F1804; Sigma-Aldrich). Secondary antibodies used were goat anti-rabbit IRdye 800CW (827-08365; LI-COR) and goat anti-mouse IRdye 680LT (925-68020; LI-COR).

### Data access

The sequencing data generated from both CBE and MagnEdit editing studies are available at ArrayExpress (E-MTAB-8742).

## Supplementary Information

## Acknowledgements

We thank David Liu for providing BE3 and University of Minnesota Center for Genome Engineering colleagues for helpful comments. These studies were supported in part by a grant from the National Cancer Institute (P01-CA234228 to RS Harris). Salary support for JL McCann was provided in part by the National Science Foundation Graduate Research Fellowship (Grant Number 00039202) and for DJ Salamango from the University of Minnesota Craniofacial Research Training (MinnCResT) program (T90-DE022732) and National Institute of Allergy and Infectious Disease K99/R00 (K99-AI147811). RS Harris is the Margaret Harvey Schering Land Grant Chair for Cancer Research, a Distinguished University McKnight Professor and an Investigator of the Howard Hughes Medical Institute.

### Author Contributions

JL McCann: conceptualization, formal analysis, validation, investigation, methodology, and writing—original draft, review, and editing.
DJ Salamango: conceptualization, formal analysis, funding acquisition, validation, investigation, methodology, and writing—review and editing.
EK Law: investigation, methodology, and writing—review and editing.
WL Brown: investigation, methodology, project administration, and writing—review and editing.
RS Harris: conceptualization, formal analysis, supervision, funding acquisition, investigation, project administration, and writing—original draft, review, and editing.

### Conflict of Interest Statement

RS Harris is a co-founder, shareholder, and consultant for ApoGen Biotechnologies Inc. The other authors have no conflicts of interest to declare.

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
