## [Reviewer comments · Life Science Alliance]

Life Science Alliance

MagnEdit - Interacting factors that recruit DNA editing enzymes to single base targets

Jennifer McCann, Daniel Salamango, Emily Law, William Brown, and Reuben Harris
DOI: <https://doi.org/10.26508/lsa.201900606>

Corresponding author(s): Reuben Harris, University of Minnesota

Review Timeline:	Submission Date:	2019-11-17
	Editorial Decision:	2019-11-18
	Revision Received:	2020-01-23
	Editorial Decision:	2020-02-11
	Revision Received:	2020-02-14
	Accepted:	2020-02-17

Scientific Editor: Andrea Leibfried

Transaction Report:

Please note that the manuscript was previously reviewed at another journal and the reports were taken into account in the decision-making process at Life Science Alliance.

Referee #1 Review

Report for Author:

McCann et al. demonstrate the development of a conceptually novel approach (on a timely subject) for cytosine base editing by fusing the nicking Cas9 (Cas9n:D10A) to various proteins that "attract" the endogenous nuclear DNA deaminase, APOBEC3B (A3B) for site-specific cytosine base editing.

After testing various constructs, the authors showed that the Cas9n fused to hnRNPK achieved the greatest degree of site-specific cytosine base editing in A3B-specific manner, as shown by use with A3B inhibitors.

Importantly, it appears to out-perform the direct fusion counterpart (A3B-Cas9n), which is thought to be even less promiscuous than the original cytosine base editor (BE3). Overall, the manuscript is straightforward and shows a proof-of-concept that a transient or "hit-and-run" approach may decrease the residence time of the DNA deaminase at the targeted site, thus decreasing the propensity to deaminate adjacent cytosine nucleotides. However, there are a few points regarding targeting/base editing efficiency that needs to be addressed before publication.

Major Points:

1. The MiSeq read is rather short, knowing that there must be additional sequence information for the surrounding positions. Granted, the deamination reaction (including unwanted adjacent mutations) is thought to occur to nucleotides within the R-loop caused by Cas9 binding DNA. However, this observation has only been shown for the covalently fused base editors in which the distance from the deaminase and the binding position is fixed. In the original BE3 paper¹, the authors tested the effect of changing linker length and composition on the base-editing window, demonstrating an optimal linker. It may be the case that the non-covalent approach impacts (extends/reduces) the base editing window. Therefore, it would be reassuring to see more of the surrounding sequencing data near the editing site.
2. The lack of sites tested is concerning. The eGFP reporter is a nice assay, but constrains the possible target sites Cas9 can bind as well as the nucleotide composition at any one site within the eGFP gene. It is highly desirable to show multiple targeted sequences for a comparison of on-target base editing efficiencies between their MagnEdit system and other cytosine base editors.

Minor points:

1. Even though the actual DNA deaminase enzyme is "untethered" the "attractant" is still a direct fusion, and any insight into their linker construction may be valuable information.
2. Including a target sequence that is rich in cytosine nucleotides within the base editing window would be quite convincing if the MagnEdit system outperforms other cytosine base editors. For example, Extended Data Figure 3 from Konor et al 2016.

Referee #2 Review

Report for Author:

General summary: The central premise of this manuscript is to use a non-covalent method, MagnEdit, to attract APOBEC3B (A3B) for precise single cytosine base editing. This study explores multiple potential A3B-interacting proteins, and demonstrates that one of these interactors, hnRNPU1, recruits A3B to edit target cytosine. The data further suggest that the MagnEdit system may lower the frequency of target-adjacent mutations. However, one of the major concerns of this study is that the data also show that the editing efficiency of MagnEdit is significantly lower than the CBE. Though authors claim that the "hit-and-run" kinetics of the MagnEdit system serve as an advantage over covalent fusion variants, they have not convincingly demonstrated that these kinetics are responsible for the study's observed effect, and the method's relatively low efficiency minimizes the added benefits of improved on-target base specificity. In addition, it is somewhat disappointing that testing the editing capability and efficiency of MagnEdit system to any endogenous target is lacking in this study.

Major points

- The data are not sufficient to show that the hnRNPU1-Cas9n can efficiently track and increase the occupancy of A3B to the target site. This raises another question whether the hnRNPU1-Cas9n specifically tracks solely A3B or if it is able to attract other proteins to the target site which may cause unexpected effects to the target site.
- Even though the authors claim that the MagnEdit system can track endogenous A3B for base

editing, there are no data to support this idea. The results in Fig. 2b suggests that this system might not work without co-overexpression of exogenous A3B. If this is due to the expression level of endogenous A3B in 293T cells, the application of MagnEdit may be restricted by cell types if reliant on the co-overexpression of A3B. Also, ectopic expression of A3B might cause unwanted mutation to the genome.

- The authors examine the on-target and target-adjacent editing by using deep sequencing in eGFP-positive pools. However, to better monitor the target-adjacent editing, I would suggest the authors to examine the eGFP-negative pools as well. In addition to this, the design of MagnEdit (hnRNAPU1L-Cas9n) is comprised of the A3B-interactor to Cas9 nickase to attract A3B for editing, which means the targeting range would be wider compared to the A3B-Cas9n. This is probably why the target-adjacent editing of MagnEdit at the -7 position (3.6%) is higher than that at the -5 position (0.9%), whereas the target-adjacent editing of A3B-Cas9n at the -7 (16%) position is lower than at the -5 position (27%). Therefore, the authors should consider examining a broader range for the target-adjacent editing of MagnEdit, as well as any potential off-target editing events genome-wide through sequencing. This would give authors some sense about the promiscuity of the MagnEdit system, as well as consequences potentially introduced by A3B overexpression.
- Since the authors are overexpressing A3B (and not merely recruiting endogenous A3B), how is this approach any better than directed evolution or rational mutagenesis of an A3B-Cas9n fusion to be more specific in editing while maintaining on-target editing levels?

Minor points

- In this study, FACS sorting is used to determine the percentage of editing. It would be good to also include the FACS sorting data and gating strategy employed to isolate the eGFP+ populations, not just the percentage bar graph.
- Could the authors explain why the "empty" and "BFP" fusion with Cas9n have higher editing percentage with gRNA than without gRNA in Fig. 1c?
- Recent publications have detailed transcriptome-wide, off-target deamination of cytosines at the RNA level that may occur with use of CBEs. Do authors have a means to show that this phenomenon is not also occurring in their MagnEdit system, and that this is not also contributing to eGFP signal restoration?

Referee #3 Review

Report for Author:

McCann et al. propose an alternate strategy for C base editing: instead of directly fusing a deaminase to Cas9, they fuse Cas9 to hnRNPU1L, which recruits APOBEC3B. They demonstrate the activity of their construct using a fluorescent C>T reporter, both episomally and chromosomally integrated, in mammalian cells. Finally, they perform HTS to confirm editing. The concept of recruiting a deaminase by fusing Cas9 to a natural deaminase binding partner is interesting and novel; and the data shows that the system works at a low level. However, there is not enough data to support many key claims. The authors speculate about off-target editing but never actually measure it, and they have not conducted sufficient experiments to support their claim of reduced target-adjacent editing. The utility and novelty of the editor are also decreased by the need to overexpress a

deaminase in trans, instead of using endogenous A3B. If MagnEdit worked (1) at a higher efficiency and (2) without requiring overexpression of a deaminase, it would be much more impactful.

Major issues

1. The authors claim that MagnEdit reduces target-adjacent editing. However, they only test one protospacer, in which only their target cytosine is in a 5'TC context, which is the preferred sequence context of A3B. To claim that reduced editing of other C's in the window is from their proposed "hit and run" mechanism as opposed to a more stringent 5' TC substrate requirement of their system, the authors absolutely must test other sites with multiple 5'TC substrates in the editing window and show that only single C's are edited in those protospacers as well.

2. The authors speculate that MagnEdit will have lower off-target editing of DNA and RNA, but never test these major claims. The MagnEdit system still requires the overexpression of a deaminase, so there is no reason to assume that it would have fewer off-target effects than deaminase fusions. Proper experiments should be done to assess these claims. To test this, the authors should minimally:

- a. Examine DNA off-targets: use an on-target gRNA that has been previously characterized, and sequence off-target loci.
- b. Generate cDNA from cells treated with different editors, then amplicon sequence highly expressed genes to search for off-target RNA editing.

3. The authors do not include all the components of proper C base editors, or use current forms of CBEs, which results in extremely low (<10%, often <2%) editing. This is a major problem, as it creates a misleading straw-man comparison with a "current CBE"-which is apparently not a current CBE but rather one that is missing a critical component, the UGI-and likely greatly reduces the editing efficiency of their MagnEdit system as well. Why is a UGI not used, as it is for nearly all other cytosine base editors? The authors should try repeating their experiments using proper UGI-containing constructs and improved-expression constructs generated by the Dow or Liu labs.

Minor criticisms:

4. In figures 1c and 2a-d, the y axes are incorrectly labeled as "% editing." They should be relabeled as "% GFP positive cells." This is important because % editing could be much lower than the % GFP positive cells, especially for the episomal reporter in which correcting of only a small fraction of reporter would lead to a GFP positive cell.

5. The authors should discuss potential side effects of overexpressing hnRNPUL1 in cells.

6. Recruiting a deaminase through accessory proteins tethered to Cas9 is not as novel as the authors imply. Bassik and coworkers and Huang and coworkers have recruited in trans deaminases through MS2 fusions and SunTagging. The difference here is that the recruitment partner binds to the deaminase itself, so recruitment does not rely on tagging the deaminase. However, this novelty is dampened because MagnEdit does not work with endogenous A3B and still requires A3B overexpression from a separate cassette. If an exogenous A3B cassette needs to be introduced anyways, there is not much benefit to overexpressing A3B as opposed to an MS2-tagged or SunTagged deaminase.

November 18, 2019

Re: Life Science Alliance manuscript #LSA-2019-00606-T

Dr. Reuben S Harris
University of Minnesota
Biochemistry, Molecular Biology and Biophysics
321 Church St. SE
6-155 Jackson Hall
Minneapolis, MN 55455

Dear Dr. Harris,

Thank you for transferring your manuscript entitled "MagnEdit - Interacting Factors that Recruit DNA Editing Enzymes to Single Base Targets" to Life Science Alliance. The manuscript was assessed by expert reviewers at another journal before, and the editors transferred those reports to us with your permission.

The reviewers who evaluated your study elsewhere raised some technical issues and thought that the advantage of the method over existing ones is somewhat limited. You already provided a response to the concerns raised upfront and a revision outline, and we concluded that such a revised version is suitable for publication here, pending that the revision indeed addresses the technical concerns. We would thus like to invite you to submit the revised manuscript to Life Science Alliance.

Thank you for this interesting contribution to Life Science Alliance. We are looking forward to receiving your revised manuscript.

Sincerely,

B. MANUSCRIPT ORGANIZATION AND FORMATTING:

Responses to Referee's Comments

Referee #1

“McCann et al. demonstrate the development of a conceptually novel approach (on a timely subject) for cytosine base editing by fusing the nicking Cas9 (Cas9n:D10A) to various proteins that "attract" the endogenous nuclear DNA deaminase, APOBEC3B (A3B) for site-specific cytosine base editing.

After testing various constructs, the authors showed that the Cas9n fused to hnRNPK achieved the greatest degree of site-specific cytosine base editing in A3B-specific manner, as shown by use with A3B inhibitors.

Importantly, it appears to out-perform the direct fusion counterpart (A3B-Cas9n), which is thought to be even less promiscuous than the original cytosine base editor (BE3). Overall, the manuscript is straightforward and shows a proof-of-concept that a transient or "hit-and-run" approach may decrease the residence time of the DNA deaminase at the targeted site, thus decreasing the propensity to deaminate adjacent cytosine nucleotides.

Response: Thank you for appreciating our main points and the novelty of our non-covalent approach. Please note that hnRNPU1 (not K) achieved the greatest degree of site-specific cytosine base editing, which is an easy mistake to make given the large number of hnRNPs.

However, there are a few points regarding targeting/base editing efficiency that needs to be addressed before publication.

Major Points:

1. The MiSeq read is rather short, knowing that there must be additional sequence information for the surrounding positions. Granted, the deamination reaction (including unwanted adjacent mutations) is thought to occur to nucleotides within the R-loop caused by Cas9 binding DNA. However, this observation has only been shown for the covalently fused base editors in which the distance from the deaminase and the binding position is fixed. In the original BE3 paper¹, the authors tested the effect of changing linker length and composition on the base-editing window, demonstrating an optimal linker. It may be the case that the non-covalent approach impacts (extends/reduces) the base editing window. Therefore, it would be reassuring to see more of the surrounding sequencing data near the editing site.

Response: As recommended, we have revised **Fig. 3** to include increased MiSeq read lengths spanning from the -22 position to the +22 position, relative to the target “C”. Please note that nucleobases beyond this length are not well resolved and fail quality control.

2. The lack of sites tested is concerning. The eGFP reporter is a nice assay, but constrains the possible target sites Cas9 can bind as well as the nucleotide composition at any one site within the eGFP gene. It is highly desirable to show multiple targeted sequences for a comparison of on-target base editing efficiencies between their MagnEdit system and other cytosine base editors.

Response: As recommended, we now include a **new Fig. 4**, which uses two different chromosomal loci - *FANCF* and *EMXI* - to further demonstrate the specificity of the MagnEdit system. These particular chromosomal targets were chosen to facilitate comparisons with prior studies that used the same sites and have shown, as reproduced here, high frequencies of adjacent

off-target editing by covalent base editing constructs such as BE3 (PMID 27096365, 28585549, 29746667).

Minor points:

1. Even though the actual DNA deaminase enzyme is "untethered" the "attractant" is still a direct fusion, and any insight into their linker construction may be valuable information.

Response: We have revised the methods to include this information.

2. Including a target sequence that is rich in cytosine nucleotides within the base editing window would be quite convincing if the MagnEdit system outperforms other cytosine base editors. For example, Extended Data Figure 3 from Komor et al 2016.

Response: Please see response #2 above.

Referee #2

General summary: The central premise of this manuscript is to use a non-covalent method, MagnEdit, to attract APOBEC3B (A3B) for precise single cytosine base editing. This study explores multiple potential A3B-interacting proteins, and demonstrates that one of these interactors, hnRNPUL1, recruits A3B to edit target cytosine. The data further suggest that the MagnEdit system may lower the frequency of target-adjacent mutations. However, one of the major concerns of this study is that the data also show that the editing efficiency of MagnEdit is significantly lower than the CBE. Though authors claim that the "hit-and-run" kinetics of the MagnEdit system serve as an advantage over covalent fusion variants, they have not convincingly demonstrated that these kinetics are responsible for the study's observed effect, and the method's relatively low efficiency minimizes the added benefits of improved on-target base specificity. In addition, it is somewhat disappointing that testing the editing capability and efficiency of MagnEdit system to any endogenous target is lacking in this study.

Response: We agree that this initial version of MagnEdit partly compromises efficiency for fidelity. We also fully agree that efficiency and fidelity are challenges for the whole field. There are two general approaches to tackle these problems – try to provide efficient editors with higher fidelity (as many groups appear to be doing) or try to make higher fidelity editors more efficient. Now that we have achieved a higher fidelity proof-of-concept with MagnEdit, we are hopeful that we and others will be able to advance this concept and make it more efficient with future studies. We are confident that our novel concept and system should be shared with the community through publication so that our group and many others can work on future improvements.

In addition and as recommended, we now include a **new Fig. 4**, which uses two different chromosomal loci - *FANCF* and *EMXI* - to further demonstrate the improved specificity of the MagnEdit system. These particular chromosomal targets were chosen to facilitate comparisons with prior studies with covalent CBEs including BE3 that used the same sites and shown, as reproduced here with A3B-Cas9n/gRNA, high frequencies of adjacent off-target editing by covalent base editing constructs (PMID 27096365, 28585549, 29746667). In contrast, the MagnEdit system yielded significantly higher rates of on-target editing (desired single C-to-T mutations) than the covalent A3B-Cas9n complex, as well as correspondingly lower rates of

target-adjacent editing.

Major points

- The data are not sufficient to show that the hnRNAPUL1-Cas9n can efficiently track and increase the occupancy of A3B to the target site. This raises another question whether the hnRNAPUL1-Cas9n specifically tracks solely A3B or if it is able to attract other proteins to the target site which may cause unexpected effects to the target site.

Response: Please see the control experiments in our manuscript and particularly in **Fig. 2A, B & C**. The results in **Fig. 2A** indicate that hnRNAPUL1 uniquely attracts A3B to the editing site. The results in **Fig. 2B** show that reporter-specific gRNA, the hnRNAPUL1-Cas9n complex, and A3B are all essential for eGFP Leu202 editing. The results in **Fig. 2C** show that only catalytically active (not inactive) A3B is capable of activating the reporter. Taken together, these data indicate that hnRNAPUL1-Cas9n specifically attracts A3B and that other cellular proteins are not an issue.

- Even though the authors claim that the MagnEdit system can track endogenous A3B for base editing, there are no data to support this idea. The results in Fig. 2b suggests that this system might not work without co-overexpression of exogenous A3B. If this is due to the expression level of endogenous A3B in 293T cells, the application of MagnEdit may be restricted by cell types if reliant on the co-overexpression of A3B. Also, ectopic expression of A3B might cause unwanted mutation to the genome.

Response: This is a fair point because not all cells express endogenous A3B (ex. PMID 20308164, 23389445). However, most cells in the human body express one or more related APOBEC3 enzymes capable of DNA deaminase activity. We are confident that the proof-of-concept studies here will encourage future studies to optimize the system for endogenous A3B as well as design similar MagnEdit systems for related APOBEC enzymes that are more broadly expressed (such as A3C), thus circumventing the need to provide an active DNA deaminase *in trans*.

- The authors examine the on-target and target-adjacent editing by using deep sequencing in eGFP-positive pools. However, to better monitor the target-adjacent editing, I would suggest the authors to examine the eGFP-negative pools as well. In addition to this, the design of MagnEdit (hnRNAPUL1-Cas9n) is comprised of the A3B-interactor to Cas9 nickase to attract A3B for editing, which means the targeting range would be wider compared to the A3B-Cas9n. This is probably why the target-adjacent editing of MagnEdit at the -7 position (3.6%) is higher than that at the -5 position (0.9%), whereas the target-adjacent editing of A3B-Cas9n at the -7 (16%) position is lower than at the -5 position (27%). Therefore, the authors should consider examining a broader range for the target-adjacent editing of MagnEdit, as well as any potential off-target editing events genome-wide through sequencing. This would give authors some sense about the promiscuity of the MagnEdit system, as well as consequences potentially introduced by A3B overexpression.

Response: These are good points and we have addressed them three different ways. First, we show results from full MiSeq reactions in a revised **Fig. 3B**. These sequences span the entire *cis*-region surrounding the eGFP editing site and show very low (near/at background) frequencies of editing at other positions in MagnEdit reactions. Second, as shown in a **new Fig. 4**, *FANCF* and *EMX1* are used as representative chromosomal editing sites and, again, MagnEdit elicits high

frequencies of on-target editing and lower frequencies of adjacent off-target events and no detectable off-target events (outside gRNA editing window) compared to current CBE methods. Third, as recommended, we also sequenced the *FANCF* and *EMXI* editing sites in GFP-negative pools (i.e. non-editing enriched pools) and found no differences from sequencing the GFP-positive pools. At present we are not showing these large data sets but can add them to the paper if you still feel it is necessary.

- Since the authors are overexpressing A3B (and not merely recruiting endogenous A3B), how is this approach any better than directed evolution or rational mutagenesis of an A3B-Cas9n fusion to be more specific in editing while maintaining on-target editing levels?

Response: As discussed in our opening response to your comments, we think it is fair to say that both approaches are encouraging and much more work will be necessary to determine the “best” methodology for highly specific and efficient editing of single cytosine bases. In the long-term, it is possible that both approaches will evolve to be more advantageous in different applications.

Minor points

- In this study, FACS sorting is used to determine the percentage of editing. It would be good to also include the FACS sorting data and gating strategy employed to isolate the eGFP+ populations, not just the percentage bar graph.

Response: We now include a workflow and representative dot plots in the composite figure on the next page. We feel that most labs (all labs with a little consultation) should be capable of FACS purification of eGFP-positive cells from an mCherry-positive pool and therefore we would prefer not to include these data in the manuscript. However, if you and the editor still feel they should be included, we can add them as Supplementary Fig. S1.

A

B

C

D

- Could the authors explain why the "empty" and "BFP" fusion with Cas9n have higher editing percentage with gRNA than without gRNA in Fig. 1c?

Response: These data indicate that single-stranded DNA in R-loops created by gRNA annealing is targeted by A3B (even without non-covalent MagnEdit or covalent guiding to the target). Indeed, in a recent manuscript from our lab, the synthetic lethal combination of A3B expression and uracil DNA glycosylase ablation (knockout or inhibition) led us to infer that R-loops are substrates for endogenous A3B deamination (PMID 31611371). **Fig. 1C** is further evidence that R-loops are naturally preferred substrates for A3B and we are currently working to elaborate on this idea.

- Recent publications have detailed transcriptome-wide, off-target deamination of cytosines at the RNA level that may occur with use of CBEs. Do authors have a means to show that this phenomenon is not also occurring in their MagnEdit system, and that this is not also contributing to eGFP signal restoration?

Response: This is a great point but beyond the scope of the present manuscript. In other studies in our lab we have been able to show that A3B is not capable of RNA editing in part due to a low catalytic efficiency and in part to an active site that will not accommodate an RNA cytosine nucleobase (PMID 30130104 and studies in process). Moreover, in a cellular system, we have been able to demonstrate that A3A is unique amongst human APOBEC3 family members with respect to RNA editing activity (Levin-Klein *et al.*, manuscript in process). We apologize for not being able to provide more information here but we hope that you appreciate that the focus of the current study is DNA (not RNA) editing.

Referee #3:

McCann *et al.* propose an alternate strategy for C base editing: instead of directly fusing a deaminase to Cas9, they fuse Cas9 to hnRNPUL1, which recruits APOBEC3B. They demonstrate the activity of their construct using a fluorescent C>T reporter, both episomally and chromosomally integrated, in mammalian cells. Finally, they perform HTS to confirm editing. The concept of recruiting a deaminase by fusing Cas9 to a natural deaminase binding partner is interesting and novel; and the data shows that the system works at a low level. However, there is not enough data to support many key claims. The authors speculate about off-target editing but never actually measure it, and they have not conducted sufficient experiments to support their claim of reduced target-adjacent editing. The utility and novelty of the editor are also decreased by the need to overexpress a deaminase in trans, instead of using endogenous A3B. If MagnEdit worked (1) at a higher efficiency and (2) without requiring overexpression of a deaminase, it would be much more impactful.

Response: These are all fantastic points and thank you very much for noting that our work is “interesting and novel”. We hope you also appreciate that this study is a proof-of-concept. Much like “base editing” started with low efficiencies and rapidly evolved into more efficient versions (BE1, 2, 3, 4, etc), we are confident that MagnEdit has similar potential for growth. Improvements in efficiency and in attracting endogenous deaminases are clearly important topics for future studies.

Major issues

1. The authors claim that MagnEdit reduces target-adjacent editing. However, they only test one protospacer, in which only their target cytosine is in a 5' TC context, which is the preferred sequence context of A3B. To claim that reduced editing of other C's in the window is from their proposed "hit and run" mechanism as opposed to a more stringent 5' TC substrate requirement of their system, the authors absolutely must test other sites with multiple 5' TC substrates in the editing window and show that only single C's are edited in those protospacers as well.

Response: To address this point, we have performed studies with MagnEditing of endogenous *FANCF* and *EMXI*, which have multiple potentially editable cytosine bases within the gRNA-targeted region described in prior studies (PMID 27096365, 28585549, 29746667). As shown in a new Fig. 4, MagnEdit is more specific at these chromosomal loci in comparison to the covalently tethered A3B-Cas9n editing complex. It is also remarkably efficient at these chromosomal sites in comparison to eGFP, which are observations we will follow-up in future studies.

2. The authors speculate that MagnEdit will have lower off-target editing of DNA and RNA, but never test these major claims. The MagnEdit system still requires the overexpression of a deaminase, so there is no reason to assume that it would have fewer off-target effects than deaminase fusions. Proper experiments should be done to assess these claims. To test this, the authors should minimally:

- a. Examine DNA off-targets: use an on-target gRNA that has been previously characterized, and sequence off-target loci.
- b. Generate cDNA from cells treated with different editors, then amplicon sequence highly expressed genes to search for off-target RNA editing.

Response: Please see our responses above as well as our response to the final comment by Reviewer #2.

3. The authors do not include all the components of proper C base editors, or use current forms of CBEs, which results in extremely low (<10%, often <2%) editing. This is a major problem, as it creates a misleading straw-man comparison with a "current CBE"-which is apparently not a current CBE but rather one that is missing a critical component, the UGI-and likely greatly reduces the editing efficiency of their MagnEdit system as well. Why is a UGI not used, as it is for nearly all other cytosine base editors? The authors should try repeating their experiments using proper UGI-containing constructs and improved-expression constructs generated by the Dow or Liu labs.

Response: We apologize for not describing our constructs clearly. All of our constructs do indeed have UGI covalently fused to the C-terminus of the Cas9 nickase (*i.e.*, all are derived directly from BE3 and are therefore isogenic apart from whatever is fused to the N-terminus of the Cas9 nickase-UGI complex). We have revised our methods accordingly and trust that the details are now clear and fully reproducible.

Minor criticisms:

4. In figures 1c and 2a-d, the y axes are incorrectly labeled as "% editing." They should be relabeled as "% GFP positive cells." This is important because % editing could be much lower than the % GFP positive cells, especially for the episomal reporter in which correcting of only a

small fraction of reporter would lead to a GFP positive cell.

Response: We have relabeled the y-axes accordingly (but note that the %editing and %eGFP+ are likely the same for the single-copy chromosomal reporter editing experiments).

5. The authors should discuss potential side effects of overexpressing hnRNPUL1 in cells.

Response: We could not find any literature indicating potential side effects of overexpressing hnRNPUL1 in cells and have included the following sentence in the first paragraph of discussion: “... Although we have not observed nor found literature indicating potential side effects of overexpressing hnRNPUL1 in cells, heterologous attractants such as single-chain antibodies may be even better for promoting the non-covalent editing of single target cytosine bases.”

6. Recruiting a deaminase through accessory proteins tethered to Cas9 is not as novel as the authors imply. Bassik and coworkers and Huang and coworkers have recruited in trans deaminases through MS2 fusions and SunTagging. The difference here is that the recruitment partner binds to the deaminase itself, so recruitment does not rely on tagging the deaminase. However, this novelty is dampened because MagnEdit does not work with endogenous A3B and still requires A3B overexpression from a separate cassette. If an exogenous A3B cassette needs to be introduced anyways, there is not much benefit to overexpressing A3B as opposed to an MS2-tagged or SunTagged deaminase.

Response: These are fair points and we have revised our discussion to include this alternative technologies.

February 11, 2020

RE: Life Science Alliance Manuscript #LSA-2019-00606-TR

Dr. Reuben S Harris
University of Minnesota
Biochemistry, Molecular Biology and Biophysics
321 Church St. SE
6-155 Jackson Hall
Minneapolis, MN 55455

Dear Dr. Harris,

Thank you for submitting your revised manuscript entitled "MagnEdit - Interacting factors that recruit DNA editing enzymes to single base targets". We have now assessed your revised manuscript and point-by-point response to the concerns of the reviewers. We appreciate the introduced changes and, importantly, the inclusion of endogenous targets in the analysis. One reviewer asked you to check off-target effects in GFP-negative cells (ie, where on-target editing obviously failed). Should your sequencing results that you are mentioning in your point-by-point response include those data on the GFPLeu202 adjacent region, we'd appreciate inclusion of this control as a supplementary file. Additionally, please pay attention to the following when preparing the final files of your manuscript:

- Please also enter the middle initials for all co-authors in our submission system
- Please add a callout in the manuscript text to Fig 4G
- Please provide the source data for the blots shown in Figure 1C
- Please make sure that others can use your method - you can use as much space as needed in the material & methods section to allow others to apply the method easily

Once we receive the final files upon this minor revision, we can swiftly move to acceptance and publication of your work. Congratulations on this very nice method!

A. FINAL FILES:

B. MANUSCRIPT ORGANIZATION AND FORMATTING:

Sincerely,

Andrea Leibfried, PhD
Executive Editor
Life Science Alliance
Meyerhofstr. 1

69117 Heidelberg, Germany
t +49 6221 8891 502
e a.leibfried@life-science-alliance.org
www.life-science-alliance.org

Subject: **Minor Revisions for manuscript #LSA-2019-00606-T**

Dear Dr. Leibfried,

Thank you for the positive decision on our manuscript titled “**MagnEdit – Interacting Factors that Recruit DNA Editing Enzymes to Single Base Targets**”. We have completed the following requested minor revisions:

- One reviewer asked you to check off-target effects in GFP-negative cells (ie, where on-target editing obviously failed). Should your sequencing results that you are mentioning in your point-by-point response include those data on the GFPLeu202 adjacent region, we'd appreciate inclusion of this control as a supplementary file.” *Response:* As recommended, we now include these results as **Supplementary Figure S1** and have added the following text to the end of the results section: “... In addition, higher *FANCF* and *EMXI* on-target editing frequencies and similar adjacent off-target trends were evident for MagnEdit versus the covalently tethered A3B-Cas9n-CBE in eGFP-negative pools (**Supplementary Fig. S1**). These additional results from sequencing the “dark” population suggested that on-target chromosomal editing events may far exceed those that yield functional correction of the *eGFP* Leu202 reporter.”
- Please also enter the middle initials for all co-authors in our submission system. *Response:* added as requested.
- Please add a callout in the manuscript text to Fig 4G. *Response:* added as requested.
- Please provide the source data for the blots shown in Figure 1C. *Response:* These blots have been included in an eps file for editorial review. As you can see the boxed regions have been used for the composite in Figure 1C and, apart from cropping for presentation, are unmodified. We are happy to provide these data in any format you wish but we do not see any reason to include them as supplementary material.
- Please make sure that others can use your method - you can use as much space as needed in the material & methods section to allow others to apply the method easily. *Response:* We have reviewed our methods and feel they are sufficiently detailed for use by other labs.

We trust that you will now find our manuscript ready for publication in *LSA*.

February 17, 2020

RE: Life Science Alliance Manuscript #LSA-2019-00606-TRR

Dr. Reuben S Harris
University of Minnesota
Biochemistry, Molecular Biology and Biophysics
321 Church St. SE
6-155 Jackson Hall
Minneapolis, MN 55455

Dear Dr. Harris,

Thank you for submitting your Methods entitled "MagnEdit - Interacting factors that recruit DNA editing enzymes to single base targets". It is a pleasure to let you know that your manuscript is now accepted for publication in Life Science Alliance. Congratulations on this interesting work.

DISTRIBUTION OF MATERIALS:

Again, congratulations on a very nice paper. I hope you found the review process to be constructive and are pleased with how the manuscript was handled editorially. We look forward to future exciting submissions from your lab.

Sincerely,
